# Investigating the role of climate-related disasters in the relationship between food insecurity and mental health for youth aged 15–24 in 142 countries

**Isobel Sharpe** **, Colleen M. Davison** *

Department of Public Health Sciences, Queen's University, Kingston, Ontario, Canada

* davisonc@queensu.ca

## Abstract

Food insecurity (FI) represents a major global health challenge. Because climate-related disasters are a determinant of both FI and poor mental health, we investigated whether the severity of these disasters intensifies the relationship between FI and youth mental health. Data on FI and mental health came from the Gallup World Poll, a nationally representative survey of individuals in 142 countries, which included 28,292 youth aged 15–24. Data on climate-related disasters came from the International Disaster Database, a country-level record of disasters. Multilevel negative binomial regression was used to calculate relative risk (RR) of poor mental health. Youth with moderate or severe FI were significantly more likely to report poor mental health experiences compared to those with none/mild FI (moderate: RR 1.37, 95% confidence interval (CI) 1.32–1.41; severe: 1.60, 95% CI 1.54–1.66). We also observed a weak yet significant interaction effect (p<0.0001), which suggested that the country-level relationship between FI and poor mental health is slightly stronger at greater disaster severity. While further research is needed to improve our understanding of these complex relationships, these findings suggest that mental health should be considered when undertaking national climate change actions and that additional FI-related supports may work to improve youth mental health.

## Introduction

Food insecurity (FI), or the lack of access to sufficient, safe, and nutritious food [1], represents a major global health challenge [2]. FI is associated not only with various negative physical health outcomes, including reduced nutrient intake, chronic disease, and premature death [3–7], but also with negative mental health outcomes [8, 9]. Individuals may experience feelings of stress, anxiety, depression, shame, and alienation due to FI itself or their FI status relative to others [9–11]. Conversely, poor mental health may contribute to FI, for example through inability to generate steady income or to cope with challenges [12–14]. Complex, and likely bidirectional, associations exist between FI and mental health.

Investigating the role of climate-related disasters in
the relationship between food insecurity and
mental health for youth aged 15–24 in 142
countries. PLOS Glob Public Health 2(9):
e0000560. https://doi.org/10.1371/journal.
pgph.0000560

AUSTRALIA

**Data Availability Statement:** The data from the
Gallup World Poll that support the findings of this
study were accessed through colleagues at McGill

University's Institute for Global Food Security, who held a partnership with the Food and Agriculture Organization. Restrictions apply to the availability of these data, which were used under license for this study. The data from the International Disaster Database that support the findings of this study are openly available at https://public.emdat.be/.

**Funding:** This study was supported by a research grant from the Canadian Institutes for Health Research PJT-162463 (CMD Co-Investigator, Frank Elgar Principal Investigator). The first author was supported by a Canadian Institutes for Health Research Fredrick Banting and Charles Best Canada Graduate Scholarship - Master's Award (IS). The funders had no role in study design, data collection and analysis, decision to publish, or preparation of the manuscript.

**Competing interests:** The authors have declared that no competing interests exist.

Youth, ages 15 to 24 years, are vulnerable with respect to FI and mental health. Youth often brings important life changes such as starting an occupation or changes in relationships and living situations, presenting potential opportunities for worsened FI status [15, 16]. This is especially true at the upper end of this age range, when young adults often become more independent from their parents [17, 18]. Many youth also struggle with mental health concerns. In 2019, approximately 86 million youth aged 15–19 were living with a mental disorder [19]. Furthermore, suicide was the fourth leading cause of death among this age group [20]. Poor mental health has the potential to affect psychological development, learning, and relationship building in youth and throughout the life course. Relatively few studies have assessed the relationship between FI and mental health among youth, particularly at the global level and using well-tested measures of FI [21, 22].

Climate change, largely driven by human greenhouse gas emissions, represents one of the most serious current threats to human health and wellbeing [23]. One particularly concerning aspect of climate change is the increase in frequency and intensity of climate-related natural disasters such as floods, storms, heatwaves, and droughts [23, 24]. Climate-related disasters are a key contextual determinant of both FI and mental health. The 2020 Global Report on Food Crises identified climate-related disaster events as the second-most common driver of acute FI, affecting 34 million people across 25 countries [25]. Climate-related disasters contribute to FI through unexpected disruptions to crop yields and to physical and economic access to food. Climate-related disasters also harm mental health through their connection to psychological trauma for populations facing these events [26–28]. The effects of climate-related disasters fall disproportionately on those populations most vulnerable [29], working to intensify existing inequities such as those related to both FI and mental health. Youth is often a time of developmental and sociodemographic change, meaning that the cascading effects of climate-related disasters, such as disruptions to school and work, economic instability, and displacement [30], may create vulnerabilities within this population.

The potential three-way relationship among climate-related disasters, FI, and mental health has not been well studied. To our knowledge, there has just been a single study with this focus. Using a nationally representative Australian sample, Friel and colleagues [31] measured the association between FI and mental health at varying levels of drought. The authors found support for the idea that drought exposures modify the association between FI and mental health, but emphasised that these relationships were complex and challenging to quantify. Notably, the study tested three binary indicators of FI and therefore may not have been accurate in capturing the overall FI construct in comparison to an internationally validated multi-component scale [31]. Another methodological issue, common within this area of literature, was that the study explored the effects of a single disaster event on a relatively small population [31, 32]. In summary, more work is needed to better understand the relationships among climate-related disasters, FI, and mental health, particularly at a global scale using validated measures.

Thus, the main objective of our study was to determine whether climate-related disasters modify the relationship between FI and poor mental health among youth globally. We were particularly interested in understanding whether the severity of climate-related disasters intensifies the FI-mental health relationship.

## Methods

### Data sources

The proposed objective was addressed using a cross-sectional design with data from two secondary sources. The first data source was the 2017 cycle of the Gallup World Poll (GWP), a nationally representative survey of individuals aged 15 and older from 148 countries [33]. In

each country, a representative sample of ~1000 individuals was collected using one of two techniques. Telephone surveys were used in those countries with at least 80% telephone coverage or where telephone surveys were customary. To sample individuals by phone, either random digit dialling or selection from a nationally representative phone number list was used. In the remainder of countries, face-to-face surveys were conducted. Individuals were selected through a stratified multi-stage cluster sampling process. In the first stage, clusters of households were stratified by population size and/or geography and 100–125 clusters were selected. In the second stage, 8–10 households were selected from each of the clusters using random route procedures. In the third stage, one respondent was randomly selected from each household. Further details on the GWP survey methodology are available elsewhere [34]. To address our specific research objective, we sampled 'youth' between ages 15–24 based on the United Nations (UN) definition [35]. For the purposes of our study, the GWP provided individual-level data on the exposure, outcome, and potential confounders of interest. Gallup obtained informed consent from all participants and the governing bodies of each country approved the survey protocols.

The second data source was the International Disaster Database (EM-DAT), a public global database of natural and technological disasters and their impacts developed by the Centre for Research on the Epidemiology of Disasters [36]. All disasters included in the database met at least one of the following criteria: 1) at least 10 people were reported killed, 2) at least 100 people were reported affected, 3) a state of emergency was declared, or 4) a call for international assistance occurred. For the current study, EM-DAT provided country-level data on the effect modifier of interest: climate-related disaster severity. Based on the GWP 2017 survey year, we chose to capture climate-related disaster severity data from 2015–2017. This three-year observation window was selected in an attempt to capture the time period through which climate-related disasters may impact individuals' FI and/or mental health in the GWP data year 2017 [37–40]. While some disaster-related effects are very immediate (e.g., local food sources destroyed, disaster-related trauma), others are more gradual (e.g., long-term disruptions to crop yield, lasting mental health problems).

### Measures

**Food insecurity.**   FI was measured using the GWP's Food Insecurity Experiences Scale (FIES) [41]. The FIES is an 8-item questionnaire that measures individual-level FI over the previous 12 months (S1 Text). The FIES is a valid and reliable psychometric measure [42–44]. Its Rasch reliability falls between 0.70–0.80 for 79% of countries included in the GWP [42]. Further, strong correlations between the FIES and related measures, such as prevalence of undernourishment (0.79) and child malnutrition (0.60), suggest its validity [43]. For the current analysis, responses to all eight FIES questions were coded as 'yes' = 1 and 'no' = 0. The responses were then summed and categorised into the following levels of FI: none or mild (score 0–3), moderate (score 4–6), or severe (score 7–8). Scores were only calculated for those who provided a valid response to each question; those with missing data or who responded 'don't know' or 'refused' to any of the eight items were counted as missing.

**Poor mental health.**   Poor mental health was measured using the GWP's Daily Experience Index (DEI) (S1 Text). The DEI measures level of wellbeing through the existence or absence of positive and negative feelings [34]. At the country level, the DEI has a Cronbach's alpha value of 0.72 [34], indicating a satisfactory level of internal consistency [45]. In the present analysis, the DEI was calculated by coding all ten individual items where an answer reflecting negative emotion received a score of 1 and all other answers received a score of 0. We summed the responses to all 10 items, resulting in a count score for poor mental health ranging from 0

to 10 (where a higher score represents worse mental health). Following Gallup's methodology [34], we did not calculate the DEI score if more than two individual items were missing ('don't know' or 'refused').

**Climate-related disasters.** Data on climate-related disasters came from the EM-DAT. Climate-related disasters were defined as those with meteorological, hydrological, or climatological origins, as described by EM-DAT's disaster classification system [36]: extreme temperatures, fog, storms, floods, landslides, wave actions, droughts, glacial lake outbursts, and wildfires.

EM-DAT reports disasters at the country level. For the present analysis, we were interested in identifying those countries whose populations may be most seriously affected by climate-related disasters and comparing them with countries whose populations are less seriously affected. Thus, we categorised each country as 'high' or 'low' according to their level of climate-related disaster severity and assigned the severity variable to all individuals within that country. Disaster severity was based on the total number of people killed per 1,000,000 population in a given country over the time period of interest. This was, therefore, a country-level sum of those presumed dead as a result of all climate-related disaster events that occurred in that country between 2015–2017. To isolate those countries most seriously affected by climate-related disasters, we categorised the top 10% of countries as high severity and the bottom 90% as low severity. We chose the 90th percentile as a cut-off value due to the exponential distribution of this variable and the obvious divide between countries with many climate-related disasters and those with far fewer at about the 90th percentile mark (S1 Text).

Missing values for number of deaths were set to 0 under the assumption that disaster events associated with high mortality were less likely to be unreported than those with few or no deaths [46]. Data from the World Bank [47] were used to adjust for country population size. As per Ward and Shively [48], we used population data from 2014 (one year before the time period of interest) to account for the fact that a country's population size may be affected by the disasters themselves.

**Potential confounders.** The following variables were selected as potential confounders based on existing literature [8, 22, 49–52] and their availability in the GWP dataset: the respondent's age (years) and gender (male, female), level of urbanicity of the respondent's home (rural or farm, small town or village, suburb of a large city, large city), number of children in household <15 years of age, marital status of the respondent (single/never married, married, separated, divorced, widowed, domestic partner), highest level of education completed by the respondent (elementary or less, secondary, tertiary), employment status (full time for an employer, full time self-employed, part time and not want to work full time, part time and want to work full time, unemployed, out of the workforce) and annual household income (international dollars). The household income variable was log transformed due to its diminishing returns on mental wellbeing [8]. Due to their small number, all 'don't know' or 'refused' responses for these variables were counted as missing.

## Statistical analysis

The final study sample included 142 countries. Six countries–Brazil, Maldives, Mauritania, Moldova, Turkmenistan, and Vietnam–were removed due to large amounts of missing data for the variables of interest. An additional 194 observations were removed due to missing outcome data. The remaining number of missing observations was proportionally small (<5%) and treated as missing at random. To accommodate for the differences in sampling design between those countries using face-to-face surveys (multi-stage stratified cluster sampling)

and those using telephone surveys (stratified random sampling), all strata with a single primary sampling unit were pooled.

Descriptive statistics were presented for the study sample, accounting for the appropriate stratification, clustering, and sampling weights. Further, poststratification weights were adopted to improve the global representativeness of the sample at the country level [8]. All descriptive statistics were presented as mean (standard error of the mean; SEM) for continuous variables and as frequency (%) for categorical variables.

Multilevel negative binomial regression models with robust standard error estimates were used to generate a relative risk (RR) and corresponding 95% confidence interval (CI) for the poor mental health outcome. The negative binomial model was used due to overdispersion in the Poisson model. All models were unweighted and fitted using SAS PROC GLIMMIX, where we included a random intercept to account for added variation at the country level. We first generated bivariate models to examine each predictor. The backwards selection technique was then used to generate a parsimonious adjusted model, with the likelihood ratio test used to assess changes in model fit. The first adjusted model tested FI as the main predictor. To better understand how the relationship between FI and poor mental health varied around the world, we stratified our findings by country income level, UN sub-region, and individual country. Country-specific results were presented on a world map using ArcGIS software by Esri. The second adjusted model evaluated the addition of an interaction between FI and climate-related disaster severity (high vs. low). Besides the two main models, we performed sensitivity analyses to test various conceptualisations of the climate-related disaster severity variable.

The regression analyses were ≥90% powered to detect a RR of 1.5 for outcomes ranging in prevalence from 20%-60%. Level of significance was set at $p < 0.05$ unless otherwise specified. All statistical analyses were conducted using SAS 9.4 (SAS Inc., Cary, North Carolina, USA).

## Results

The final study sample from the 2017 GWP survey consisted of 28,292 youth from 142 countries (S1 Text). The characteristics of the study sample are presented in Table 1. Based on FIES scores, 77.8% of youth reported none or mild FI, 11.2% reported moderate FI, and 11.0% reported severe FI. The mean score for poor mental health was 2.29 (SEM 0.06) out of 10. S1 Text presents a histogram of poor mental health scores, showing that the majority of youth scored low (≤2) on this measure. During the three-year period leading up to the GWP survey year (2015–2017) a total of 828 climate-related disasters occurred within the 142 countries of interest, resulting in 27,508 reported deaths (S1 Text). Of those disasters, 47.7% were floods, 33.2% were storms, 6.6% were landslides, 4.5% were wildfires, 4.1% were droughts, and 3.9% were extreme temperatures. The countries with the highest total number of people killed per 1,000,000 population over the 2015–2017 period were Sierra Leone (158.5 per 1,000,000), Haiti (61.1 per 1,000,000), France (49.9 per 1,000,000), Belgium (36.8 per 1,00,000), and Sri Lanka (26.1 per 1,000,000).

Table 2 shows the results of the multilevel negative binomial regression models examining the association between FI and poor mental health with country as a random effect. Compared to the bivariate models, most of the fully adjusted RR estimates from the multivariate model were slightly attenuated but remained consistent in their relative size and direction. Notably, FI produced the strongest effect of all predictors in the multivariate model. Youth with moderate or severe FI were significantly more likely to report experiences of poor mental health compared to those with none or mild FI (moderate FI: RR 1.37, 95% CI 1.32–1.41; severe FI: 1.60, 95% CI 1.54–1.66). In addition, experiences of poor mental health were significantly predicted by older age (RR 1.02, 95% CI 1.02–1.03), greater number of children in the household (RR

**Table 1. Characteristics of the 2017 Gallup World Poll study sample (n = 28,292 youth from n = 142 countries).**
Descriptive estimates were based on weighted data.

| Characteristic | Weighted n | % |
|---|---|---|
| Food insecurity (FIES score) | | |
| None or mild (0–3) | 21,014 | 77.8 |
| Moderate (4–6) | 3,028 | 11.2 |
| Severe (7–8) | 2,974 | 11.0 |
| Missing | 1,422 | |
| Gender | | |
| Male | 14,665 | 51.7 |
| Female | 13,709 | 48.3 |
| Urbanicity | | |
| Rural | 11,071 | 39.1 |
| Small town | 9,562 | 33.8 |
| Suburban | 2,325 | 8.2 |
| Urban | 5,353 | 18.9 |
| Missing | 63 | |
| Marital status | | |
| Single | 21,669 | 76.5 |
| Married | 6,073 | 21.4 |
| Separated | 60 | 0.2 |
| Divorced | 100 | 0.4 |
| Widowed | 51 | 0.2 |
| Domestic partner | 366 | 1.3 |
| Missing | 56 | |
| Education (highest level completed) | | |
| Elementary or less | 9,271 | 32.7 |
| Secondary | 17,242 | 60.9 |
| Tertiary | 1,818 | 6.4 |
| Missing | 43 | |
| Employment | | |
| Full-time (employer) | 6,340 | 22.3 |
| Full-time (self-employed) | 2,846 | 10.0 |
| Part-time (seeking full-time) | 2,455 | 8.7 |
| Part-time (not seeking full-time) | 1,201 | 4.2 |
| Unemployed | 2,140 | 7.5 |
| Out of workforce | 13,393 | 47.2 |
| | **Mean** | **SEM** |
| Age (years) | 19.53 | 0.08 |
| Number of children in household aged <15 years | 1.09 | 0.03 |
| Missing (weighted n) | 30 | |
| Poor mental health (DEI score) | 2.29 | 0.06 |
| Annual household income (international $, thousands) | 17.64 | 2.80 |
| Log annual household income (international $) | 8.90 | 0.05 |

Notes: Used the stratification, clustering, and weighting variables from the GWP dataset. Also applied poststratification weights (Gallup's survey weights multiplied by country population size).

Abbreviations: Daily Experience Index (DEI), Food Insecurity Experiences Scale (FIES), standard error of the mean (SEM).

**Table 2. Bivariate and multivariate multilevel negative binomial regression models examining the association between food insecurity (Food Insecurity Experiences Scale) and poor mental health (Daily Experience Index score) among n = 28,292 youth from the 2017 Gallup World Poll survey.**

| Factor | Bivariate Models | | Multivariate Model | |
|---|---|---|---|---|
| | RR (95% CI) | p | RR (95% CI) | p |
| Food insecurity | | | | |
| None or mild | 1 (ref) | | 1 (ref) | |
| Moderate | 1.43 (1.39–1.48) | <0.0001 | 1.37 (1.32–1.41) | <0.0001 |
| Severe | 1.70 (1.63–1.77) | <0.0001 | 1.60 (1.54–1.66) | <0.0001 |
| Gender | | | | |
| Male | 1 (ref) | | 1 (ref) | |
| Female | 1.03 (1.00–1.06) | 0.0270 | 1.03 (1.00–1.05) | 0.0690 |
| Urbanicity | | | | |
| Urban | 1 (ref) | | 1 (ref) | |
| Suburban | 1.00 (0.96–1.04) | 0.9522 | 0.99 (0.95–1.03) | 0.6755 |
| Small village | 1.01 (0.98–1.04) | 0.3482 | 0.97 (0.95–1.00) | 0.0527 |
| Rural | 1.08 (1.04–1.13) | <0.0001 | 1.01 (0.97–1.05) | 0.7062 |
| Marital status | | | | |
| Single | 1 (ref) | | 1 (ref) | |
| Domestic partner | 1.14 (1.08–1.21) | <0.0001 | 1.03 (0.98–1.08) | 0.2743 |
| Married | 1.17 (1.12–1.22) | <0.0001 | 1.06 (1.01–1.10) | 0.0092 |
| Separated | 1.38 (1.24–1.53) | <0.0001 | 1.14 (1.02–1.28) | 0.0177 |
| Divorced | 1.27 (1.13–1.42) | <0.0001 | 1.13 (0.99–1.29) | 0.0621 |
| Widowed | 1.49 (1.28–1.73) | <0.0001 | 1.34 (1.13–1.58) | 0.0005 |
| Education | | | | |
| Tertiary | 1 (ref) | | 1 (ref) | |
| Secondary | 1.01 (0.97–1.06) | 0.6235 | 1.03 (0.98–1.09) | 0.1779 |
| Elementary or less | 1.09 (1.03–1.15) | 0.0042 | 1.06 (1.00–1.13) | 0.0423 |
| Employment | | | | |
| Full-time (employer) | 1 (ref) | | 1 (ref) | |
| Full-time (self-employed) | 0.99 (0.94–1.04) | 0.6305 | 0.97 (0.93–1.02) | 0.2670 |
| Part-time (seeking full-time) | 0.94 (0.90–0.99) | 0.0127 | 0.94 (0.90–0.98) | 0.0062 |
| Part-time (not seeking full-time) | 0.93 (0.89–0.97) | 0.0008 | 0.95 (0.91–0.99) | 0.0197 |
| Unemployed | 1.06 (1.01–1.11) | 0.0100 | 1.01 (0.97–1.06) | 0.5327 |
| Out of workforce | 0.83 (0.80–0.87) | <0.0001 | 0.87 (0.83–0.91) | <0.0001 |
| Number of children in household <15 years | 1.01 (1.01–1.02) | <0.0001 | 1.01 (1.00–1.01) | 0.0035 |
| Age | 1.03 (1.03–1.04) | <0.0001 | 1.02 (1.02–1.03) | <0.0001 |
| Log annual household income | 0.92 (0.91–0.93) | <0.0001 | 0.96 (0.95–0.97) | <0.0001 |
| Disaster severity[a] | | | | |
| Low | 1 (ref) | | 1 (ref) | |
| High | 1.13 (0.99–1.30) | 0.0801 | 1.05 (0.93–1.19) | 0.4217 |

Notes: Models present relative risk (RR) estimates with 95% confidence intervals (CI). All models were adjusted for clustering at the country level (random effect).

[a]Disaster severity was measured at the country level. High disaster severity was defined as those countries in the top 10% of total number of climate-related disaster deaths per 1,000,000 population between the years of 2015–2017. Low disaster severity was defined as those countries in the bottom 90% of total number of climate-related disaster deaths per 1,000,000 population between the years of 2015–2017.

1.01, 95% CI 1.00–1.01), and lower annual household income (RR 0.96, 95% CI 0.95–0.97). Those who were educated at an elementary level or lower were significantly more likely to report poor mental health experiences compared to those with a tertiary level education (RR 1.06, 95% CI 1.00–1.13). Youth who were widowed (RR 1.34, 95% CI 1.13–1.58), separated

(RR 1.14, 95% CI 1.02–1.28), or married (RR 1.06, 95% CI 1.01–1.10) were significantly more likely to report experiences of poor mental health compared to those who were single. Further, those working part-time (seeking full-time hours: RR 0.94, 95% CI 0.90–0.98; not seeking full-time hours: RR 0.95, 95% CI 0.91–0.99) or who were out of the workforce (RR 0.87, 95% CI 0.83–0.91) were significantly less likely to report poor mental health experiences compared to those working full-time hours.

We performed a series of stratified analyses to determine whether the relationship between FI and poor mental health varied across different areas of the world. First, we stratified the multivariate model by country income level (S1 Text). We observed a consistent positive association between FI and poor mental health among youth across low-, middle-, and high-income levels, with RR values ranging from 1.31–1.45 for moderate FI and from 1.53–1.74 for severe FI (vs. none or mild FI). Second, we stratified the model by UN sub-region (Fig 1). In all 19 sub-regions, youth with moderate and/or severe FI were significantly more likely to report poor mental health experiences compared to those with none or mild FI. Third, we stratified the model by individual country (Fig 2). The strongest positive associations between FI and poor mental health among youth were largely concentrated in the Western countries, including Canada, the United States, Australia, Germany, the United Kingdom, and much of Eastern Europe.

To determine whether climate-related disasters modify the relationship between FI and mental health among youth, we added a FI*disaster severity interaction term to the multivariate model from Table 2. This term was not statistically significant (interaction p = 0.5362; S1

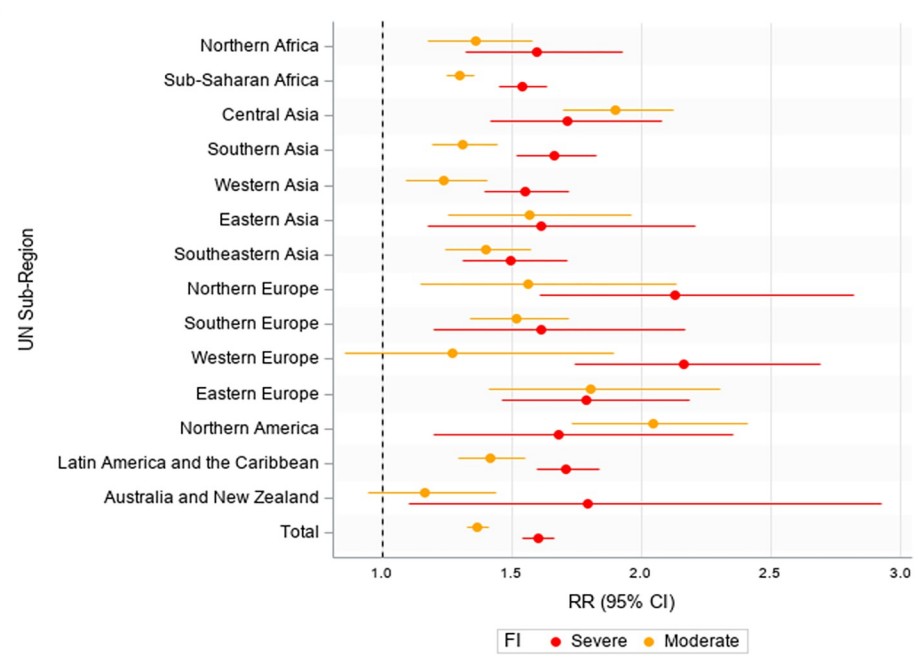

**Fig 1. Forest plot showing the association between food insecurity (Food Insecurity Experiences Scale) and poor mental health (Daily Experience Index score) among n = 28,292 youth from the 2017 Gallup World Poll survey, by United Nations sub-region.** Notes: There are 19 UN sub-regions (https://unstats.un.org/unsd/methodology/m49/overview/). Estimates were adjusted for gender, urbanicity, marital status, education, employment, number of children <15 in household, age, log annual household income, disaster severity (fixed effects), and country (random effect). Abbreviations: United Nations (UN), relative risk (RR), confidence interval (CI), food insecurity (FI).

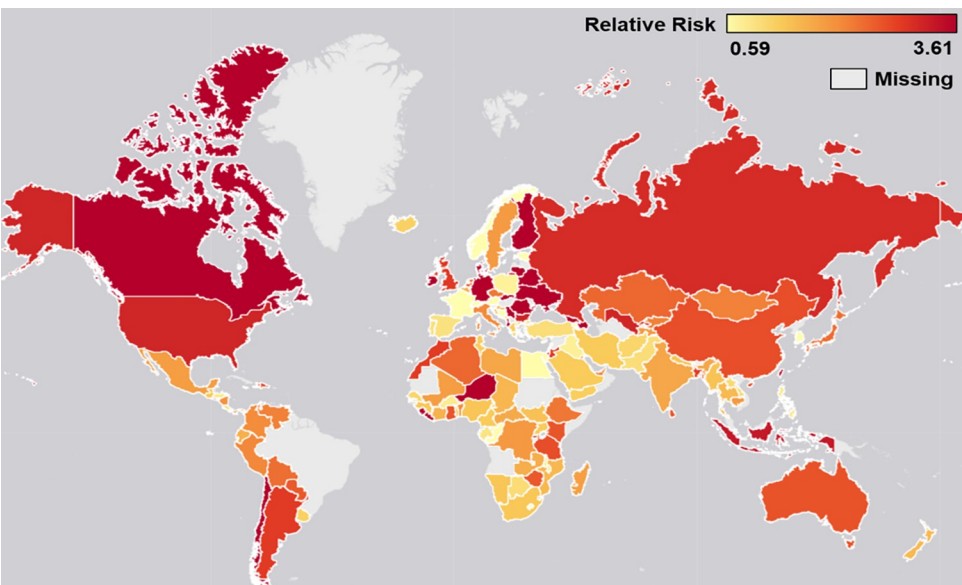

**Fig 2. Heat map depicting the strength of association between food insecurity (Food Insecurity Experiences Scale) and poor mental health (Daily Experience Index score) in n = 137 countries.** Notes: Presents relative risk estimates for moderate or severe FI vs. none or mild FI. Estimates were adjusted for gender, urbanicity, marital status, education, employment, number of children <15 in household, age, and log annual household income (fixed effects). Five of the 142 countries in the Gallup World Poll sample were not included in the map (the regression models for Hungary, Latvia, North Macedonia, and Slovenia did not converge, Kosovo was not available in the map template). Base map available at: https://www.arcgis.com/home/item.html?id=ee8678f599f64ec0a8ffbfd5c429c896 [53].

Text) and model fit did not improve according to the likelihood ratio test. Fig 3 (left) provides a graphical representation of the interaction, showing that while poor mental health increased with greater FI, this effect did not differ by level of disaster severity.

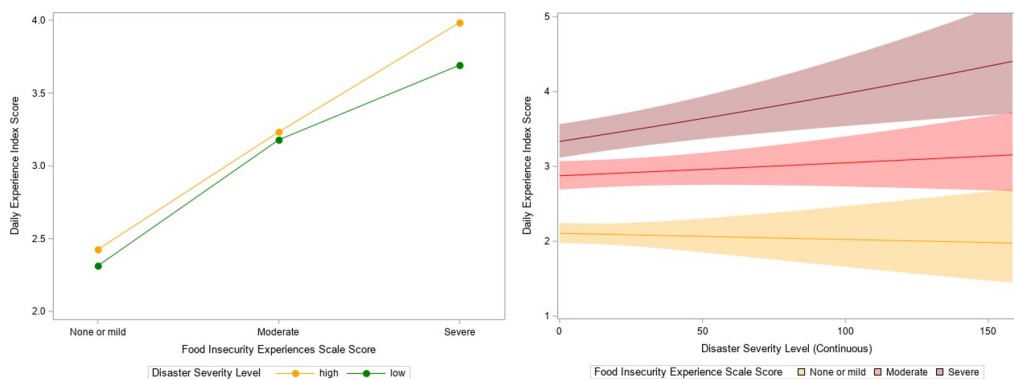

**Fig 3. Interaction plots showing the marginal effect of food insecurity (Food Insecurity Experiences Scale) and disaster severity on poor mental health (Daily Experience Index score) among n = 28,292 youth from the 2017 Gallup World Poll survey.** Left: disaster severity represented as a dichotomous variable[a]. Right: disaster severity represented as a continuous variable with 95% confidence intervals[b]. Notes: Adjusted for gender, urbanicity, marital status, education, employment, number of children <15 in household, age, log annual household income, disaster severity (fixed effects), and country (random effect). [a]Disaster severity was measured at the country level. High disaster severity was defined as those countries in the top 10% of total number of climate-related disaster deaths per 1,000,000 population between the years of 2015–2017. Low disaster severity was defined as those countries in the bottom 90% of total number of climate-related disaster deaths per 1,000,000 population between the years of 2015–2017. [b]Disaster severity was measured at the country level, defined as the total number of climate-related disaster deaths per 1,000,000 population between the years of 2015–2017 (continuous).

## Sensitivity analyses

We performed three sensitivity analyses to address some of the limitations associated with our conceptualisation of the climate-related disaster severity variable. First, we tested disaster severity as a continuous measure of disaster deaths, as opposed to its initial representation as a binary (high vs. low) variable. In doing so, we observed a very weak yet significant effect of the FI*disaster severity interaction on poor mental health (interaction p<0.0001). Table 3 displays the results of this model, showing that the interaction was significant for severe FI (p = 0.0063) but not moderate FI (p = 0.1817) when compared to none or mild FI. These findings are also reflected in the interaction plot (Fig 3, right), which shows a very small increase in poor mental health with increasing disaster severity for the severe FI group (thus a more steeply sloped line) compared to none or mild FI group. Second, we tested disaster severity as a continuous frequency measure, capturing the total number of climate-related disasters per 1,000,000 km$^2$ country surface area. In this case, the FI*disaster severity interaction was not statistically significant (interaction p = 0.1098; S1 Text). Lastly, we changed the time period of interest for the disaster severity variable from 2015–2017 to 2017 only. Again, the FI*disaster severity interaction was not statistically significant (interaction p = 0.1923; S1 Text).

## Discussion

Youth are vulnerable with respect to both FI and mental health. Within this context, they are susceptible to the additional burdens caused by climate-related disasters such as floods, storms, and heatwaves. Through our multilevel analysis, we explored whether the context of climate-related disaster severity significantly worsened the FI-mental health relationship among this age group. Few studies have explored the relationship among these three variables to date, particularly at a global level.

Our analysis revealed two main findings. For one, we observed a significant association between FI and poor mental health among youth globally. This result was in line with several previous analyses of GWP data, conducted among both youth [21, 22] and adults [8, 49–52]. Our analysis controlled for a number of indicators of poverty and material deprivation, such as annual household income, education, and employment status. This suggests that the relationship between FI and mental health among youth is not solely driven by poverty and deprivation, indicating that other psychosocial or biological mechanisms are likely also at play [22]. For example, individuals may develop what is known as toxic stress, a condition that occurs under chronic pressures, such as prolonged experiences of FI, and lack of adequate supports [54–56]. Toxic stress is has been implicated in poor mental health in the context of children and youth [56–58]. Additionally, those experiencing FI tend to consume inexpensive foods of poor nutritional quality and thereby are at a greater risk of malnutrition [7]. Malnutrition itself can contribute to poor mental health through reduced function of the brain and gut microbiome [59, 60].

Figs 1 and 2 show that the association between FI and poor mental health is potentially stronger among youth in Western countries such as Canada, the United States, Australia, and much of Eastern Europe. This finding aligns with previous work showing that the relationship between FI and poor mental health is stronger in higher-income countries and weaker in lower-income countries [50]. Experiences of FI tend to be more normalised in lower-income countries, and therefore its effect on one's mental health may not be as pronounced in comparison to the higher-income countries where FI is less common [50]. In addition, many of these countries rely on collectivistic cultures, which have been linked to greater levels of connection and support [61, 62], therefore potentially mitigating the negative mental health impacts of FI. Individuals in lower-income countries may also regularly face not only FI but also many other

**Table 3. Multivariate multilevel negative binomial regression model examining the association between food insecurity (Food Insecurity Experiences Scale) and poor mental health (Daily Experience Index score) among n = 28,292 youth from the 2017 Gallup World Poll survey.**

| | Multivariate Model | |
|---|---|---|
| **Factor** | *RR (95% CI)* | *p* |
| Food insecurity | | |
| None or mild | 1 (ref) | |
| Moderate | 1.36 (1.32–1.41) | <0.0001 |
| Severe | 1.58 (1.52–1.65) | <0.0001 |
| Gender | | |
| Male | 1 (ref) | |
| Female | 1.03 (1.00–1.05) | 0.0714 |
| Urbanicity | | |
| Urban | 1 (ref) | |
| Suburban | 0.99 (0.95–1.03) | 0.6260 |
| Small village | 0.97 (0.95–1.00) | 0.0486 |
| Rural | 1.01 (0.97–1.05) | 0.7181 |
| Marital status | | |
| Single | 1 (ref) | |
| Married | 1.06 (1.01–1.102) | 0.0089 |
| Separated | 1.15 (1.03–1.28) | 0.0141 |
| Divorced | 1.13 (0.99–1.28) | 0.0677 |
| Widowed | 1.34 (1.13–1.58) | 0.0005 |
| Domestic partner | 1.03 (0.98–1.08) | 0.2488 |
| Education | | |
| Tertiary | 1 (ref) | |
| Secondary | 1.03 (0.98–1.09) | 0.1762 |
| Elementary or less | 1.06 (1.00–1.13) | 0.0436 |
| Employment | | |
| Full-time (employer) | 1 (ref) | |
| Full-time (self-employed) | 0.97 (0.93–1.02) | 0.2841 |
| Part-time (seeking full-time) | 0.94 (0.90–0.98) | 0.0062 |
| Part-time (not seeking full-time) | 0.95 (0.91–0.99) | 0.0206 |
| Unemployed | 1.01 (0.97–1.06) | 0.5286 |
| Out of workforce | 0.87 (0.83–0.91) | <0.0001 |
| Number of children in household <15 years | 1.01 (1.00–1.01) | 0.0034 |
| Age | 1.02 (1.02–1.03) | <0.0001 |
| Log annual household income | 0.96 (0.95–0.97) | <0.0001 |
| Disaster severity[a] | 1.00 (1.00–1.00) | 0.3042 |
| Food insecurity*disaster severity[b] | | |
| None or mild*disaster severity | 1 (ref) | |
| Moderate*disaster severity | 1.00 (1.00–1.00) | 0.1817 |
| Severe *disaster severity | 1.00 (1.00–1.00) | 0.0063 |

Notes: Model presents relative risk (RR) estimates with 95% confidence intervals (CI). Model was adjusted for clustering at the country level (random effect).

[a]Disaster severity was measured at the country level. It was represented as a continuous variable: total number of climate-related disaster deaths per 1,000,000 population between the years of 2015–2017.

[b]The interaction between food insecurity and disaster severity was statistically significant (p<0.0001).

large-scale stressors, including poverty, conflict, and insecurity, meaning that FI itself likely plays a small role in overall mental health.

The nature of the observed association between FI and mental health suggests the potential for a causal relationship. Firstly, our models showed that FI had the strongest relationship with mental health of all included covariates, indicating its relative importance in predicting mental health. Similarly, Frongillo et al. [50] found that FI produced the largest standardised regression coefficient after adjusting for various economic and social development indicators such as gross domestic product and income inequality. Second, in line with previous analyses, we observed a dose-response relationship between FI and mental health at the global scale. Jones et al. [52] suggested two mechanisms through which this dose-response relationship may manifest; greater levels of FI may amplify existing psychological stressors and/or invoke multiple new pathways that compound with existing stressors to harm mental health. There is a large body of evidence linking experiences of hunger, a defining feature of severe FI as measured by the FIES [41], to serious mental health problems among children and youth, including depression and suicide-related outcomes [63–65]. Physiologically, hunger may heighten the body's response to stress in the hypothalamic-pituitary-adrenal axis, leading to subsequent negative mental health outcomes [63]. Depression and suicide-related outcomes potentially represent the pathways invoked through more severe cases of FI. Third, we found that the FI-mental health association was highly consistent. The association remained significant when the sample was stratified by country income level and UN sub-region. In their analysis of FI, state fragility, and mental health among youth, Elgar et al. [22] also reported significant associations between FI and poor mental health across all UN sub-regions. These findings suggest that the FI-mental health relationship persists among youth across cultural contexts.

Our second major finding was that while climate-related disaster severity did not modify the relationship between FI and mental health when treated as a dichotomous variable, we did see significant effect modification when it was treated as a continuous variable. Specifically, we observed a very weak yet significant interaction between FI and disaster severity in our sensitivity analysis where disaster severity was represented as a continuous measure of deaths per 1,000,000 population. This finding suggests that at the country level, the relationship between youth FI and poor mental health may grow stronger with higher levels of disaster severity. It is well known that FI negatively affects mental health through worries and shame related to food as well as through the negative mental health effects of malnutrition [8, 9]. Climate-related disasters are also directly associated with poor mental health [26–28]. Therefore, youth experiencing both climate-related disasters and FI may have compounding burdens on their mental health. Although our findings suggest that an interaction may exist, it is important to acknowledge that this effect was very weak and may not be practically meaningful. Ultimately, future research is needed to better understand and quantify this effect and the pathways through which it may operate.

## Strengths and limitations

Our study had several strengths. For one, to our knowledge it was the first of its kind to explore the relationship among youth FI, mental health, and climate-related disasters. Another strength was the use of the GWP dataset, which provided nationally representative data from 142 countries. While lack of a representative sample is a common limitation in the study of climate-related disasters, the GWP dataset allowed us to conduct a representative analysis on the global scale. Lastly, we used the FIES, a valid and reliable scale designed to provide a globally relevant measure of FI.

This analysis also had several limitations. For one, the cross-sectional nature of the data prevented us from identifying true causal relationships. It is possible that poor mental health itself may lead to FI, for example through the inability to hold steady employment or to cope with challenging situations [12–14]. As this study used pre-existing datasets, there were likely several unmeasured confounders. For example, the GWP did not report on pre-existing mental health conditions, which may predict both FI and current mental health status. Further, measures of both FI and poor mental health were self-reported, introducing the potential for social desirability bias. We also made assumptions regarding the use of EM-DAT mortality data. Foremost, the EM-DAT provided country-level data only, therefore we assigned the same value of disaster severity to all individuals within a given country. Climate-related disasters are often localised, meaning that our measure of disaster severity may not have reflected the true experiences of youth included in our study sample. To avoid ecological fallacy [66], we interpreted the effects of the disaster severity variable exclusively at the country level; however, future research should aim to collect this information at the individual level.

## Conclusions

Using data from the Gallup World Poll, we conducted a novel global analysis investigating the role of climate-related disaster severity in the relationship between food insecurity and poor mental health among 28,292 youth in 142 countries. In line with previous research, we observed a significant dose-response association between FI and poor mental health that transcended geographic and cultural contexts. Policies aimed at reducing levels of FI, such as promoting diversification in food production, improving infrastructure for those transporting and accessing food, promoting domestic production and food fortification, and reducing poverty and income inequality [2], may also be broadly effective in in improving youth mental health globally. In addition, while we observed a significant interaction effect where contexts of higher disaster severity strengthened the relationship between FI and poor mental health, its magnitude of effect was very weak. Our study provides an important starting point for understanding the pathways through which climate-related disasters and FI may synergistically harm mental health, yet further research is needed to improve our understanding of this complex system and ultimately support policy decisions.

## Supporting information

**S1 Text. Supporting information for methods and results.**
(DOCX)

## Acknowledgments

The authors would like to thank Dr. Frank Elgar of the Institute for Health and Social Policy at McGill University for providing thoughtful comments on an earlier draft of this manuscript and for assistance in gaining access to the Gallop World Poll data.

## Author Contributions

**Conceptualization:** Colleen M. Davison.

**Data curation:** Isobel Sharpe.

**Formal analysis:** Isobel Sharpe.

**Funding acquisition:** Colleen M. Davison.

**Project administration:** Isobel Sharpe.

**Supervision:** Colleen M. Davison.

**Writing – original draft:** Isobel Sharpe.

**Writing – review & editing:** Isobel Sharpe, Colleen M. Davison.

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
