## [Decision Letter · Decision Letter 0]

15 Mar 2022

PGPH-D-22-00135

Investigating the role of climate-related disasters in the relationship between food insecurity and mental health for youth aged 15-24 in 142 countries

Dear Dr. Davison,

Thank you for submitting your manuscript to PLOS Global Public Health. After careful consideration, we invite you to submit a revised version of the manuscript that addresses the points raised during the review process.

We look forward to receiving your revised manuscript.

Kind regards,

Ying Zhang, PhD, MD

Academic Editor

Journal Requirements:

1. Please provide us with a direct link to the base layer of the map used in Figure 2 and ensure this location is also included in the figure legend.

Please note that, because all PLOS articles are published under a CC BY license (creativecommons.org/licenses/by/4.0/), we cannot publish proprietary maps such as Google Maps, Mapquest or other copyrighted maps. If your map was obtained from a copyrighted source please amend the figure so that the base map used is from an openly available source.

Please note that only the following CC BY licences are compatible with PLOS licence: CC BY 4.0, CC BY 2.0  and CC BY 3.0, meanwhile such licences as CC BY-ND 3.0 and others are not compatible due to additional restrictions. If you are unsure whether you can use a map or not, please do reach out and we will be able to help you. 

The following websites are good examples of where you can source open access or public domain maps:

2. Please update your Competing Interests statement. If you have no competing interests to declare, please state: “The authors have declared that no competing interests exist.”

Additional Editor Comments (if provided):

Thank you very much for submitting your manuscript to PLOS Global Public Health.  This is an extremely important topic, and one that is of great interest to the Journal and its readers.

Having completed the evaluation of your manuscript, the reviewers recommended revisions before it can be accepted for publication outlined below. I invite you to resubmit your manuscript after addressing the comments.

When revising your manuscript, please consider all issues mentioned in the reviewers' comments carefully: please outline every change made in response to their comments and provide suitable rebuttals for any comments not addressed.

Reviewers' comments:

Reviewer's Responses to Questions

**Comments to the Author**

1. Does this manuscript meet PLOS Global Public Health’s publication criteria? Is the manuscript technically sound, and do the data support the conclusions? The manuscript must describe methodologically and ethically rigorous research with conclusions that are appropriately drawn based on the data presented.

Reviewer #1: Yes

Reviewer #2: Yes

2. Has the statistical analysis been performed appropriately and rigorously?

Reviewer #1: Yes

Reviewer #2: Yes

3. Have the authors made all data underlying the findings in their manuscript fully available (please refer to the Data Availability Statement at the start of the manuscript PDF file)?

Reviewer #1: Yes

Reviewer #2: Yes

4. Is the manuscript presented in an intelligible fashion and written in standard English?

Reviewer #1: Yes

Reviewer #2: Yes

5. Review Comments to the Author

Reviewer #1: This paper examines if the severity of climate disasters intensifies the relationship between food insecurity and mental health for youth. This is an important topic to explore given youth are the emerging future and they are experiencing more intense climatic disasters.

The authors have posed an interesting and relevant question and have used reliable data and established methods and drawn appropriate conclusions. However, the engagement with the relevant bodies of literature (food security, mental health and climate disasters) needs to be further developed. The relationship between FI and mental health lacks reference to seminal work (Radimer et al., 1992) and to the physiological outcomes FI have upon mental health and vice versa, although the authors note these in the discussion indicating they are aware of these. There is a growing literature upon the impacts of climate disasters upon mental health that can also be engaged with.

Lacking in the introduction is the concept of food security (four pillars: availability, access, utilization, stability) yet the authors mention availability (line 408) and allude to access (line 365) in the discussion. Including this concept would demonstrate the authors have engaged with this body of literature.

The authors state youth are vulnerable to FI and climate disasters but have not established youth are more vulnerable than any other age group. Some authors argue youth are the best positioned to manage during FI see comment below.

Line 46: a more current reference is the FAO et al., 2020 The state of food security and nutrition in the World 2021.

Line 47: giving some examples of negative physical health outcomes will help to balance out this paragraph and can help link to the examples given in the discussion. This is referenced this in the discussion (line 409).

Line 49: In the first introductory paragraph Radimer should be included, this work was the first to highlight the relationship between FI and feelings of worry and stress and is the first FIES question (Radimer, K. L., Olson, C. M., Greene, J. C., Campbell, C. C., & Habicht, J.-P. (1992). Understanding hunger and developing indicators to assess it in women and children. Journal of Nutrition Education, 24(1), 36S-44S. https://doi.org/10.1016/S0022-3182(12)80137-3)

Lines 50-57: states “youth are vulnerable to poor mental health” add some statistical evidence to support this as you have suggested in the discussion (line 392).This relationship would be strengthened by giving some examples: e.g. in US 2018 suicide was the second leading cause of death for youth (15-24yrs) https://www.cdc.gov/injury/wisqars/pdf/leading_causes_of_death_by_age_group_2018-508.pdf

Lines 50 -57: states “youth are vulnerable to FI” but this needs to be developed more, other studies find youth to be the least vulnerable to FI as they can between the homes of friends and extended families for meals, they can get part time work and have access to school food programs and food banks unlike preschool children who are confined to food sources at home. The paragraph really focuses upon mental health, there needs to be something on the physiological impacts of FI and youth. This relationship is mentioned in the discussion (lines 365-368) and would strengthen the paper to provide some examples in the introduction.

The second paragraph (Lines 50-57) explains only one direction how FI impacts mental health, it does not discuss the impact of poor mental health has upon FI. For example, when people are depressed and anxious it can impact their ability to gain and maintain employment and therefore access to food or have the energy to get and cook for themselves which can cause FI. This is mentioned in the strengths and limitations (line 427).

Line 69: The authors focus upon youth but have not identified any unique vulnerabilities of this population with regards to climate related disasters. Doing this would strengthen the relevance of the paper.

Methods

The methods section is well documented. The sources and measures are established.

Line 123: The FIES was developed by the FAO, you have the reference in the discussion (45) move it up here.

Results

Tables were clearly presented.

Figures were good, clear and appropriate. Figure 2 the heat map was a good illustration

Discussion

348 -349: the case has not been made that youth are particularly vulnerable to FI and climate related disasters. When the introduction is vised add how they are vulnerable.

Line 364: please explain what “toxic stress” is and why it is relevant to the condition of FI and to youth.

Line 367-368: There are three forms of malnutrition including under nutrition and over nutrition not all have shown a causal relationship to mental health and gut microbiome. The references used only consider mood, I would suggest either modify to “can contribute” or return to the literature.

Line 377: Give some examples of large scale stressors and places. It is also important to recognize that many lower-income countries operate from a communal paradigm not an individual one as found in many higher income countries and perhaps that is also a contributing factor to better mental health as youth a sense of belonging and purpose.

Line 392: in regards to the ‘dose-response’ there is an absence of physiological impacts hunger and malnutrition have upon the development of youth and rates of suicide and depression and these need to be recognized.

Line 408: unless the concept of food security is included remove “availability” as shame and worried related to FI are in response to the pillar of access – money. Availability is the production side of food security not how people gain access to the food available.

Line 418: add “youth” to the sentence.

Reviewer #2: I thank editor for the opportunity to review this paper. I thank authors for their wonderful effort looking at the interaction effect of climate related disasters in relationship with FI and mental health. Overall the manuscript has been written well, I have few minor comments on the methodology.

1. The rationale for focusing on young adults? why not adults?

2. How the consent have been drawn from the young adults?

3. How the author drawn sample size calculation? What is the rationale behind selecting 28292 sampling data?

4. The author has not described the validity and accuracy of the study tool used to assess for FI and mental health status

5. The variable of interest is drawn from the group level, how does the author justify the effect size ?

6. PLOS authors have the option to publish the peer review history of their article (what does this mean?). If published, this will include your full peer review and any attached files.

**Do you want your identity to be public for this peer review?** For information about this choice, including consent withdrawal, please see our Privacy Policy.

Reviewer #1: **Yes: **Kerrie Pickering

Reviewer #2: No

---

## [Decision Letter · Decision Letter 1]

8 Jun 2022

PGPH-D-22-00135R1

Investigating the role of climate-related disasters in the relationship between food insecurity and mental health for youth aged 15-24 in 142 countries

Dear Dr. Davison,

Thank you for submitting your manuscript to PLOS Global Public Health. After careful consideration, we feel that it has merit but meeds some minor revisions to fully meet PLOS Global Public Health’s publication criteria as it currently stands. Therefore, we invite you to submit a revised version of the manuscript that addresses the points raised during the review process.

We look forward to receiving your revised manuscript.

Kind regards,

Ying Zhang, PhD, MD

Academic Editor

Journal Requirements:

Additional Editor Comments (if provided):

I thank the authors for the efforts in making the revision, which addressed the concerns of the reviewers. Please revise again according to the reviewers' remaining comments, especially from review 2 who raised some new concerns and requested more details of the methods and justification. I understand the study was based on the Gallup World Poll data. It would be helpful to make it clearer about what methods were developed by the Gallup and what methods were decided by the authors. 

Reviewers' comments:

Reviewer's Responses to Questions

**Comments to the Author**

1. If the authors have adequately addressed your comments raised in a previous round of review and you feel that this manuscript is now acceptable for publication, you may indicate that here to bypass the “Comments to the Author” section, enter your conflict of interest statement in the “Confidential to Editor” section, and submit your "Accept" recommendation.

Reviewer #1: (No Response)

Reviewer #2: (No Response)

2. Does this manuscript meet PLOS Global Public Health’s publication criteria? Is the manuscript technically sound, and do the data support the conclusions? The manuscript must describe methodologically and ethically rigorous research with conclusions that are appropriately drawn based on the data presented.

Reviewer #1: Yes

Reviewer #2: Yes

3. Has the statistical analysis been performed appropriately and rigorously?

Reviewer #1: Yes

Reviewer #2: No

4. Have the authors made all data underlying the findings in their manuscript fully available (please refer to the Data Availability Statement at the start of the manuscript PDF file)?

Reviewer #1: Yes

Reviewer #2: Yes

5. Is the manuscript presented in an intelligible fashion and written in standard English?

Reviewer #1: Yes

Reviewer #2: Yes

6. Review Comments to the Author

Reviewer #1: Thank you for the opportunity to re-review this paper. I thank the authors for revising the manuscript to more clearly identify the mental health and food insecurity vulnerability youth may experience. In the introduction the addition of line 67-68 is important but does require further elaboration in another sentence or two to explain the bidirectional relationship.

Reviewer #2: The detailed comments for the paper has been attached in this and I would request authors to justify the issues raised clearly.

I am highlighting the major comments here.

1. The sample size calculation for drawing the total samples is required

2. The tool used for the exposure and outcome assessment has not been clearly demonstrated in terms of data collection.

3. The rational for doing binomial negative regression model

4. List of confounders are not accurate

5. Discussion has to be updated after highlighting the key study outcomes. The current part of discussion seems messy. I would request author to re work on the discussion.

7. PLOS authors have the option to publish the peer review history of their article (what does this mean?). If published, this will include your full peer review and any attached files.

**Do you want your identity to be public for this peer review?** For information about this choice, including consent withdrawal, please see our Privacy Policy.

Reviewer #1: **Yes: **Kerrie Pickering

Reviewer #2: No

---

## [Editor Report · Decision Letter 2]

1 Aug 2022

Investigating the role of climate-related disasters in the relationship between food insecurity and mental health for youth aged 15-24 in 142 countries

PGPH-D-22-00135R2

Dear Dr. Davison,

We are pleased to inform you that your manuscript 'Investigating the role of climate-related disasters in the relationship between food insecurity and mental health for youth aged 15-24 in 142 countries' has been provisionally accepted for publication in PLOS Global Public Health.

Best regards,

Ying Zhang, PhD, MD

Academic Editor